# eNEMAL, an enhancer RNA transcribed from a distal *MALAT1* enhancer, promotes NEAT1 long isoform expression

Joshua K. Stone[1]☯, Lana Vukadin[2]☯, Eun-Young Erin Ahn[2,3]*

1 Mitchell Cancer Institute, University of South Alabama, Mobile, Alabama, United States of America,
2 Division of Cellular and Molecular Pathology, Department of Pathology, University of Alabama at Birmingham, Birmingham, Alabama, United States of America, 3 O'Neal Comprehensive Cancer Center, University of Alabama at Birmingham, Birmingham, Alabama, United States of America

☯ These authors contributed equally to this work.
* eyahn@uabmc.edu

**Data Availability Statement:** The final eNEMAL sequence identified in this paper is available in GenBank under accession number MT773342.

**Funding:** This work was supported by the NIH grants (R01CA190688, R21CA185818, and

## Abstract

Emerging evidence has shown that active enhancers are abundantly transcribed, generating long non-coding RNAs, called enhancer RNAs (eRNAs). While putative eRNAs are often observed from RNA sequencing, the roles of most eRNAs remain largely unknown. Previously, we identified putative enhancer regions at the *MALAT1* locus that form chromatin-chromatin interactions under hypoxia, and one of these enhancers is located about 30 kb downstream of the *NEAT1* gene and -20 kb upstream of the *MALAT1* gene (*MALAT1*–20 kb enhancer). Here, we report that a novel eRNA, named eRNA of the NEAT1-MALAT1-Locus (eNEMAL), is transcribed from the *MALAT1*–20 kb enhancer and conserved in primates. We found that eNEMAL is upregulated in response to hypoxia in multiple breast cancer cell lines, but not in non-tumorigenic MCF10A cells. Overexpression and knockdown of eNEMAL revealed that alteration of eNEMAL level does not affect MALAT1 expression. Instead, we found that eNEMAL upregulates the long isoform of NEAT1 (NEAT1_2) without increasing the total NEAT1 transcript level in MCF7 breast cancer cells, suggesting that eNEMAL has a repressive effect on the 3'-end polyadenylation process required for generating the short isoform of NEAT1 (NEAT1_1). Altogether, we demonstrated that an eRNA transcribed from a *MALAT1* enhancer regulates NEAT1 isoform expression, implicating the *MALAT1*–20 kb enhancer and its transcript eNEMAL in co-regulation of MALAT1 and NEAT1 in response to hypoxia in breast cancer cells.

## Introduction

Most described somatic mutations in cancer are found in noncoding regions of the genome and are often contained within regulatory regions such as enhancers [1], implicating dysregulation of enhancer functions in human diseases. Enhancers are traditionally identified through multiple factors, including distance from known genes, high levels of histone-3 lysine-4 monomethylation (H3K4me1) and histone-3 lysine-27 acetylation (H3K27Ac), low levels of histone-

R01CA236911 to E.E.A.) and the internal support
from the University of Alabama at Birmingham
School of Medicine, Department of Pathology, and
O'Neal Comprehensive Cancer Center (to E.E.A).
The funders had no role in study design, data
collection and analysis, decision to publish, or
preparation of the manuscript.

**Competing interests:** The authors have declared
that no competing interests exist.

3 lysine-4 tri-methylation (H3K4me3), RNA polymerase II binding, p300/CBP binding, and
DNase I hypersensitivity sites. An emerging characteristic of active enhancers is transcription
of noncoding RNAs called enhancer RNAs (eRNAs) [2]. Activated enhancers recruit RNA
polymerase II which transcribes the enhancer to generate eRNAs. The expression level of
eRNAs serves as a functional readout of the enhancer and correlates with the expression levels
of adjacent coding genes [3]. eRNAs can be transcribed in a bidirectional manner, between
500–2,000 nucleotides in length, unspliced, and rarely polyadenylated (<5%) [3–5]. There is a
growing recognition in the role eRNAs play in chromatin looping of adjacent genomic regions
and transcriptional activation/repression of neighboring gene promoters, but mechanistic vali-
dation of the target genes of eRNAs is still lagging.

Metastasis-associated lung adenocarcinoma transcript 1 (MALAT1) and nuclear-enriched
abundant transcript 1 (NEAT1) are neighboring long noncoding RNAs (lncRNAs) separated
by less than 60 kb on human chromosome 11q13.1 [6]. Both MALAT1 and NEAT1 are abun-
dant in the nucleus and localize at nuclear speckles and paraspeckles, respectively. The *NEAT1*
gene is transcribed into two isoforms, canonically polyadenylated short transcript NEAT1_1
(~3.7 knt) and non-polyadenylated long transcript NEAT1_2 (~23 knt) [6, 7]. Previous reports
demonstrate that NEAT1_2 is essential for paraspeckle assembly [7–9]. Interestingly, NEAT1
and MALAT1 are two of the most highly upregulated lncRNAs in response to hypoxia [10].
Both MALAT1 and NEAT1 are overexpressed in many solid tumors, and play roles in tran-
scription and alternative splicing, and/or sponging microRNAs, which result in upregulation
of genes involved in cell growth, angiogenesis, invasion, and migration [11, 12].

Previously, we identified three novel *MALAT1* enhancers that form distinct long-range
chromatin-chromatin interactions with each other and the *MALAT1* promoter in breast can-
cer cells under normoxic and/or hypoxic conditions, which are not observed in non-tumori-
genic cells [13]. Two of these novel enhancers, found at -7 kb and -20 kb upstream of the
*MALAT1* transcription start site, are distally located between the *NEAT1* and *MALAT1* loci.

In this study, we report that the -20 kb enhancer is transcribed to an eRNA. We identified
that this eRNA is 762 nt in length, polyadenylated, and upregulated by hypoxia. We named it
eRNA of the NEAT1-MALAT1-Locus (eNEMAL). Surprisingly, we found that eNEMAL regu-
lates NEAT1 expression by modulating levels of the short and long isoforms in MCF7 breast
cancer cells. Therefore, we propose that eNEMAL is functional and plays a novel role in regu-
lating NEAT1 isoform selection.

## Results

### RNA sequencing data suggest the *MALAT1*–20 kb enhancer expresses a noncoding RNA

Since transcription of noncoding, enhancer RNAs (eRNAs) is now recognized as a sign of
active enhancers [14–16], we examined whether one or more of the *MALAT1* enhancers we
previously identified in the *MALAT1* locus (-20 kb enhancer, -7 kb enhancer, and 3' enhancer)
[13] may encode eRNAs which would be upregulated in response to hypoxia. We first exam-
ined the data generated by multiple RNA sequencing techniques to see if any potential
enhancer areas showed evidence of transcription. We analyzed the data from conventional
RNA-seq in conjunction with cap analysis of gene expression (CAGE), an RNA-seq variant
designed to precisely identify transcription start sites of nascent RNAs [17]. This bioinformatic
analysis demonstrated strong expression of the *MALAT1* and upstream *NEAT1* genes as mea-
sured by RNA reads on the plus strand (Fig 1A). The 3' enhancer and -7 kb enhancers did not
show any evidence of transcription. However, at the -20 kb enhancer, we observed RNA-seq
signal indicating transcription of the minus strand in the MCF7 breast cancer cell line, but not

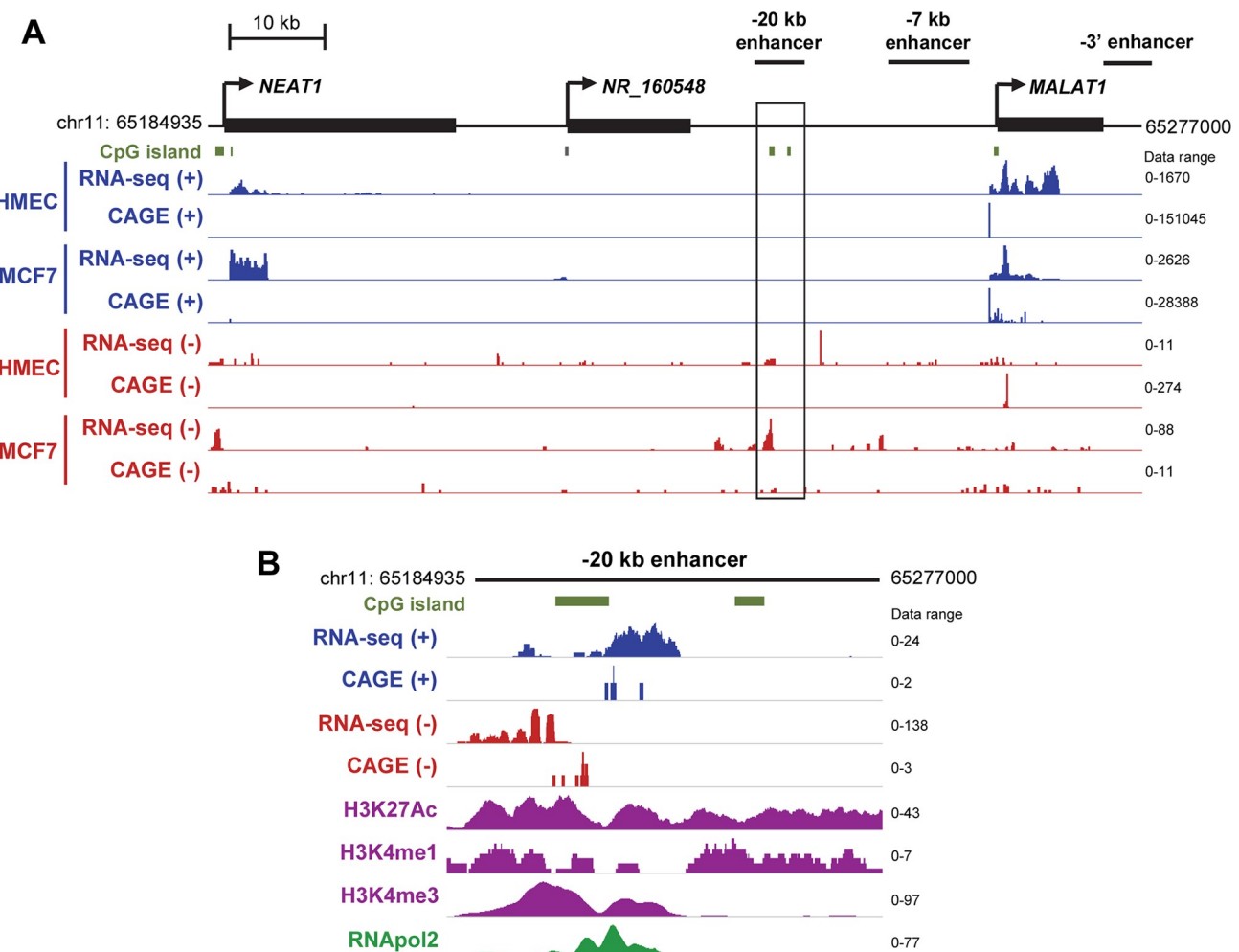

**Fig 1. The *MALAT1*–20 kb enhancer is actively transcribed in breast cancer cells.** (**A**) Bulk RNA-seq and CAGE data for normal (HMEC) and cancerous (MCF7) breast cell lines. Normalized tag counts are indicated for each row, with blue tracks corresponding to plus strand signal and red tracks corresponding to negative strand signal. RNA-seq and CAGE data were downloaded from ENCODE. (**B**) Close-up view of the *MALAT1*–20 kb enhancer in MCF7 cells. Note that different data ranges were set for each mark to display the close-up view of the peaks.

in normal human mammary epithelial cells (HMECs) or from the plus strand (Fig 1A). CAGE supported transcription initiating on the minus strand approximately -20.3 kb upstream of the *MALAT1* transcription start site, which overlapped with other epigenetic marks consistent with enhancer elements (Fig 1B). These sequencing platforms provide support for active transcription of the -20 kb enhancer in breast cancer cells.

## Characterization of the *MALAT1*–20 kb eRNA and defining it as eNEMAL

To determine the approximate size of the putative eRNA, we developed a tiling assay based upon the RNA-seq read length between -21.3 and -20 kb upstream of the *MALAT1* transcription start site, with primer pairs approximately every 150–200 bp apart (Fig 2A). This tiling assay revealed high expression from -21 to -20.3 kb, with no detectable expression up or downstream of these loci in MCF7 breast cancer cells (Fig 2A), suggesting this eRNA is transcribed from a 700 bp region.

We then used rapid amplification of cDNA ends (RACE) to determine the exact 5' transcription start site and 3' transcription termination site (Fig 2B). Sequencing of the PCR

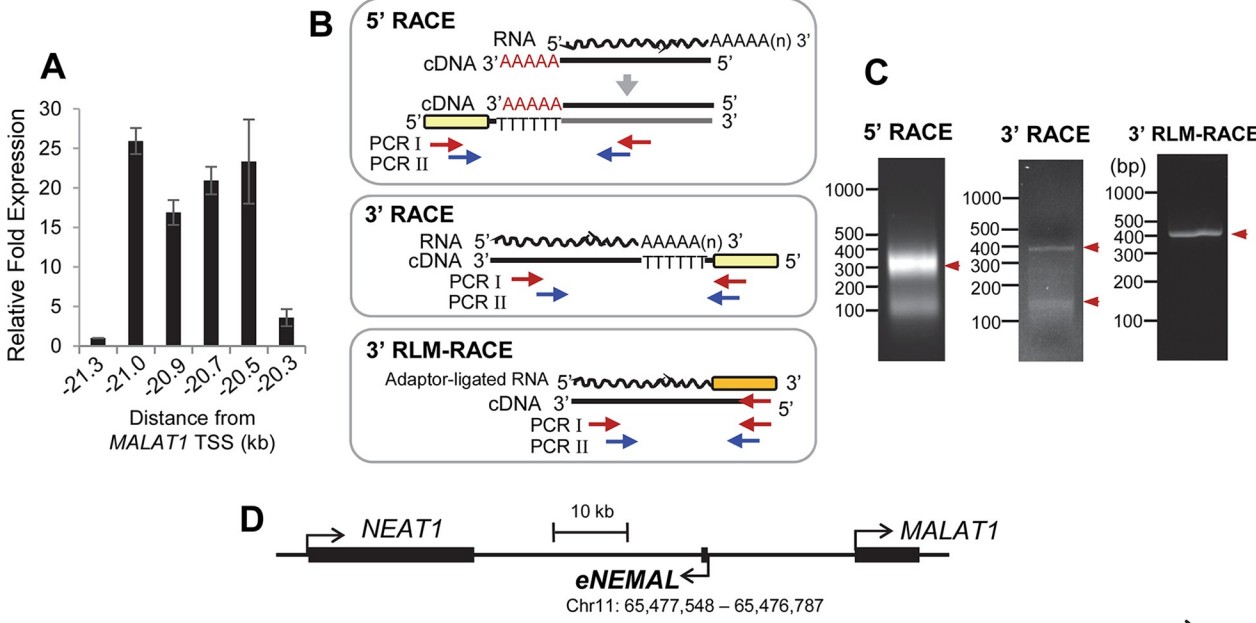

**Fig 2. eNEMAL is a polyadenylated and unspliced eRNA transcribed from the *MALAT1* -20kb enhancer.** (**A**) RT-qPCR tiling assay for MCF7 endogenous eRNA expression with coordinates relative to the *MALAT1* transcription start site. Data shown as mean ± SD, n = 3. (**B**) Schematics of 5' RACE, 3' RACE, and RNA ligase-mediated 3' RACE (3' RLM-RACE), and the locations of the primers used. RNA used for RACE was isolated from MCF7 cells after 24 hours hypoxic exposure. Red arrows, primer sets used for the first PCR; Blue arrows, primer sets used for the second PCR; Yellow bar, adaptor; Orange bar, oligonucleotide ligated to RNA. In 5'RACE, AAAAA in red font indicates the DNA sequences added by the terminal transferase reaction. (**C**) Gel images of 5' RACE, 3' RACE and 3' RLM-RACE products. Bands indicated with red arrowheads were cut and sequenced. (**D**) Genomic location of the identified eRNA, *eNEMAL*, relative to the *MALAT1* and *NEAT1* gene bodies. The position of the *eNEMAL* gene was determined using hg38 as reference genome. (**E**) Full sequence of the *eNEMAL* gene. Blue arrow downstream of the poly(A) signal indicates the poly(A)-tailed 3' end of the eNEMAL transcript determined by 3' RACE. Orange arrow indicates a non-poly(A)-tailed alternative 3' end determined by 3' RLM-RACE.

products from 5' RACE and conventional 3' RACE using oligo(dT)-containing-adaptor (Fig 2C) revealed that the eRNA was 3' polyadenylated and contains a canonical poly(A) signal 17 bases upstream of the transcription termination site (S1 Fig). Overlap between RACE primer sets and amplicons allowed us to determine that the -20 kb enhancer RNA was 762 nt in length, unspliced, and polyadenylated (Fig 2D and 2E). We also performed RNA ligase-mediated 3' RACE (3' RLM-RACE) to determine whether any non-polyadenylated transcripts are expressed from this locus (Fig 2B and 2C). From this approach, we detected a different 3' end (Fig 2E and S2 Fig). However, it is not clear that whether this non-polyadenylated transcript is another major form of this eRNA or it is a read-through product that is missed by the 3' end

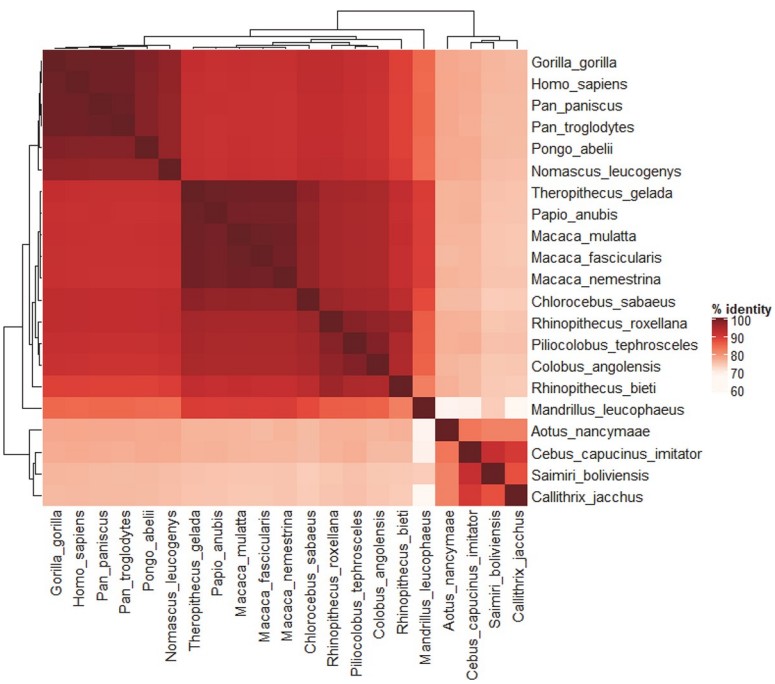

**Fig 3. eNEMAL is conserved in primates.** Orthologs of the human eNEMAL sequence were queried by BLAST and BLAT searches using NCBI and Ensembl chordate nucleotide databases. Chromosomes or contigs that hold the *MALAT1* locus were retrieved and aligned. Results with 50% identity over at least 50% of sequence length were considered for orthologs; the only sequences which satisfied these conditions were from primates. Sequences were aligned by MEGA X MUSCLE algorithm and accession numbers for each sequence are provided in S3 Fig.

cleavage and polyadenylation machinery. Therefore, we focused the polyadenylated form of the eRNA which was the only confirmative product detected by the conventional 3' RACE (S1 Fig). Taken together, these data demonstrated that the -20 kb enhancer which forms long-range chromatin interactions at the *MALAT1* locus is transcriptionally active, generating an eRNA in breast cancer cells.

Bioinformatic prediction using CPAT (Coding-Potential Assessment Tool; [18]) did not provide any evidence for coding potential in this eRNA (Ficket score, 0.6118; Hexamer score, -0.1851; Coding probability, 0.002861). Since this *eRNA* gene is found 30.5 kb downstream of *NEAT1* and 20.3 kb upstream of *MALAT1* (Fig 2E), we named it eRNA of the NEAT1-MALAT1-Locus (eNEMAL hereafter). As the synteny of the NEAT1-MALAT1 region is highly conserved in vertebrates [19], we then searched for eRNA orthologs and found the eRNA sequence is restricted to simian primates only and is highly conserved (S3 Fig). eNEMAL sequence identity ranges from 74% with *Callithrix jacchus* to 99.6% with *Gorilla gorilla* and overall sequence homology was reflective of primate evolution (Fig 3 and S1 Table).

## eNEMAL is upregulated in breast cancer cells in response to hypoxia

Using RNA-seq and CAGE data, we found that eNEMAL was more highly expressed in MCF7 breast cancer cells compared to normal HMECs (Fig 1). We then asked if eRNA upregulation was consistent in different breast cancer subtypes or if variation existed. We found that compared to the immortalized, non-tumorigenic MCF10A cell line, endogenous, normoxic levels of eNEMAL were lower in the triple negative cell lines MDA-MB-157 and MDA-MB-231, luminal A cell line MCF7, luminal B cell line MDA-MB-361, and HER2-enriched cell line MDA-MB-453 (Fig 4A). Endogenous expression was greater in the luminal A cell line T47D, luminal B cell line

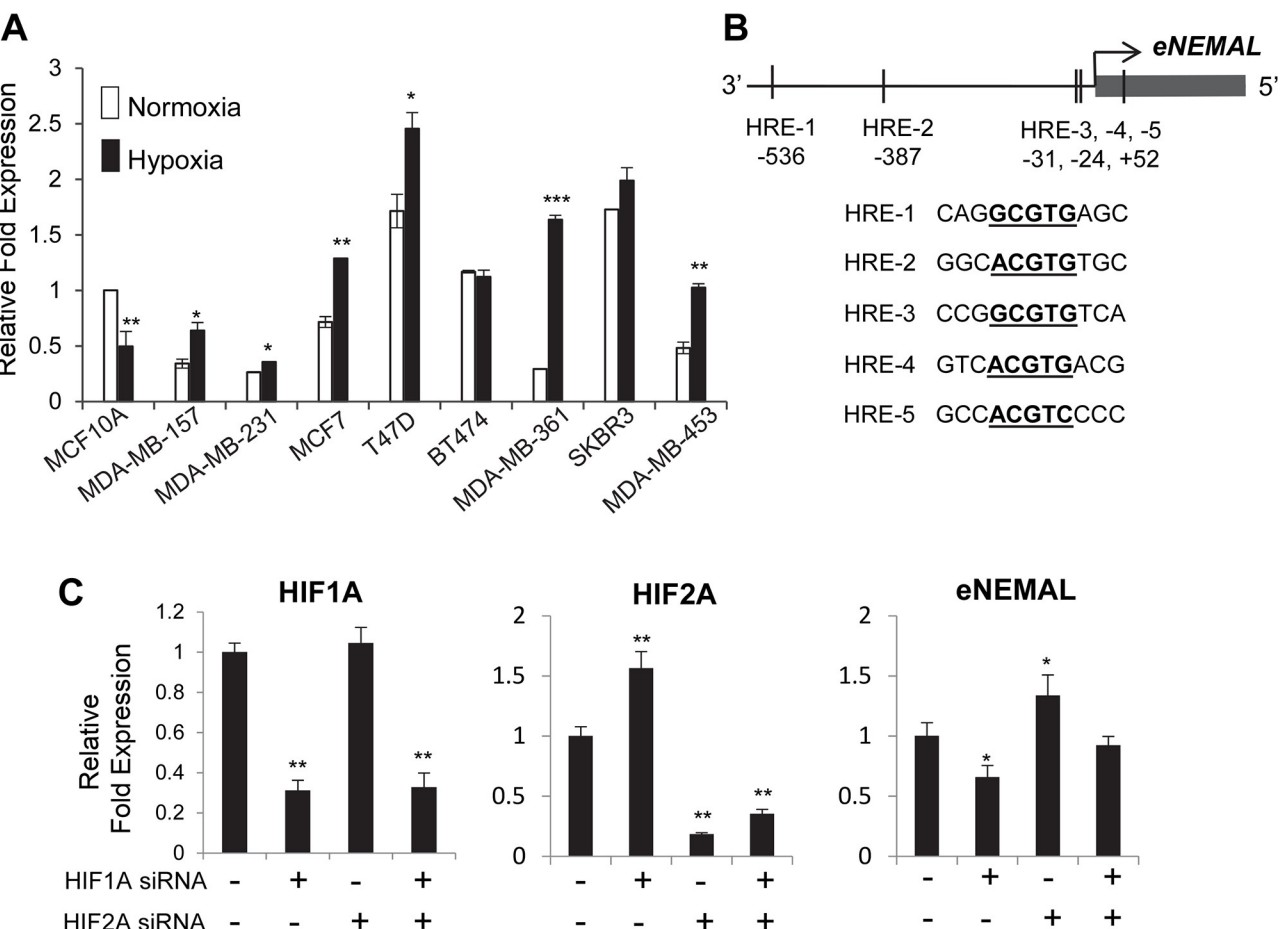

**Fig 4. eNEMAL is upregulated under hypoxia in breast cancer cells.** (**A**) eRNA expression determined by RT-qPCR following in cells grown under normoxic (21% $O_2$) and hypoxic (1% $O_2$) conditions for 24 hours, normalized to MCF10A normoxic expression. MCF10A, non-tumorigenic immortalized; MDA-MB-157 and MDA-MB-231, triple-negative; MCF7 and T47D, luminal A; BT474 and MDA-MB-361, triple positive; SKBR3 and MDA-MB-453, HER2-enriched. (**B**) Hypoxic response elements (HREs) in the *eNEMAL* promoter. (**C**) MCF7 cells were transfected with siRNA targeting HIF1A or HIF2A for 48 hours then maintained under hypoxia for additional 24 hours. The expression levels of HIF1A, HIF2A and eNEMAL were determined by qPCR. Note that we observed HIF2A upregulation upon HIF1A siRNA transfection, which is consistent with the previous finding of a repressive role of HIF1A on HIF2A expression [20]. Data shown as mean ± SD, n = 3, $^*$ $P < 0.05$, $^{**}$ $P < 0.01$, $^{***}$ $P < 0.001$.

BT474, and HER2-enriched cell line SKBR3. This indicated that the eNEMAL is variably expressed in breast cancer subtypes and has no consistent pattern under normoxic conditions.

Enhancer RNA expression is often directly correlated with that of flanking genes [3, 21–23]. As MALAT1 and NEAT1 are consistently upregulated under hypoxia in these breast cancer cell lines [10, 13, 24], we asked if hypoxia could upregulate eNEMAL as well. When the cells were incubated under hypoxia (1% $O_2$) for 24 hours, VEGFA and CA9, well-known hypoxic responsive genes, were clearly upregulated in all cell lines tested (S4 Fig). Interestingly, in non-tumorigenic MCF10A cells, eNEMAL expression was reduced under hypoxia while BT474 and SKBR3 cells did not show any changes (Fig 4A). In contrast, eNEMAL was upregulated in response to hypoxia all other cancer cell lines (Fig 4A), demonstrating that this enhancer is hypoxic-responsive in breast cancer and its upregulation under hypoxia corresponds with that of MALAT1.

The hypoxia-inducible factor (HIF) family of transcription factors predominantly activate the gene expression program in response to hypoxic stress [25, 26]. We asked if a HIF protein

could also play a role in transcription of eNEMAL. Examination of the *eNEMAL* promoter region revealed a total of five hypoxic response elements (HREs), the consensus binding motifs of the HIF family (Fig 4B). We knocked down HIF-1α and HIF-2α in MCF7 cells with siRNA under hypoxic conditions and measured change in eNEMAL expression. This reveals HIF-1α activates eNEMAL transcription but surprisingly HIF-2α represses transcription (Fig 4C). Together, these data confirm the *MALAT1*–20 kb enhancer is active under hypoxia and eRNA transcription from this enhancer is promoted by HIF-1 α.

## eNEMAL regulates NEAT1 isoform expression in MCF7 breast cancer cells

Having demonstrated eNEMAL was upregulated in response to hypoxia, we asked if this eRNA played an activating role in MALAT1 expression. We designed siRNA targeting two regions of eNEMAL and transfected MCF7 cells under hypoxia. We were able to reduce eNE-MAL expression approximately 50% by siRNA in MCF7 cells, but observed no effect on MALAT1 expression (Fig 5A). Similarly and for further confirmation of our results, we used GapmeR to reduce eNEMAL, which also showed no effect on MALAT1 expression (Fig 5B).

The half-lives of eRNAs are often quite short and are a direct read-out of enhancer activity at any given time [14, 27]. In order to confirm our knockdown results and determine the effect of a more stable eRNA pool, we cloned eNEMAL into the lentiviral vector pCDH-CMV-MCS-Puro-EF1 and examined the sole effect of eNEMAL overexpression under normoxia. The eRNA was robustly overexpressed when transfected into MCF7 cells, and MALAT1 had a slight, but statistically significant, increase in expression (Fig 5C). These results demonstrated eNEMAL overexpression alone has only a minor role in MALAT1 upregulation.

The *NEAT1* gene is located approximately 53 kb upstream of *MALAT1* (Fig 2D) and produces one of two isoforms; a short 3.7 knt isoform (NEAT1_1) from an alternative polyadenylation site or a long 23 knt isoform (NEAT1_2) (Fig 5D). While NEAT1_2 is essential for paraspeckle assembly, NEAT1_1 can contribute to paraspeckle formation only when NEAT1_2 is present [8, 9].

To determine if eNEMAL affects NEAT1 expression, we designed a primer set for NEAT1 RT-qPCR. As the short isoform NEAT1_1 sequence is completely contained within the long isoform NEAT1_2, specific primers for NEAT1_1 could not be designed. Therefore, we instead measured total NEAT1 (NEAT1_1 and NEAT1_2) using a primer set targeting both isoforms, and the NEAT1_2 expression level using a primer set specifically targeting 3' of NEAT1_2 (Fig 5D). First, we measured the NEAT1 expression level following siRNA-mediated eRNA knockdown under hypoxia. We could not detect notable changes in the total NEAT1 expression level upon eNEMAL knockdown (Fig 5E). In contrast, we found a sharp decrease in the level of NEAT1_2 transcript by two different siRNAs (Fig 5E). GapmeR-mediated eNEMAL knockdown also showed similar results (Fig 5F). Then, when we overexpressed eNEMAL, total NEAT1 was slightly downregulated (Fig 5G), indicating that eNEMAL overexpression alone does not activate NEAT1 transcription. In contrast, we observed the opposite effect with NEAT1_2 which was significantly upregulated (Fig 5G), suggesting that eNEMAL regulates the NEAT1 isoform switch in MCF7 breast cancer cells. Furthermore, we observed that in several breast cancer cell lines, total NEAT1 and NEAT1_2 were not proportionally increased upon hypoxia, and the breast cancer cell lines with high levels of eNEMAL expression (Fig 4A) generally showed a bigger increase of the NEAT1_2 portion compared to the total NEAT1 increase upon hypoxia (S5 Fig). It has been shown that NEAT1_2 production is favored when 3'-end processing (canonical polyadenylation) of the nascent NEAT1 transcript is inhibited [9]. Our data suggest that eNEMAL may have an inhibitory effect on 3' end processing and polyadenylation of NEAT1_1.

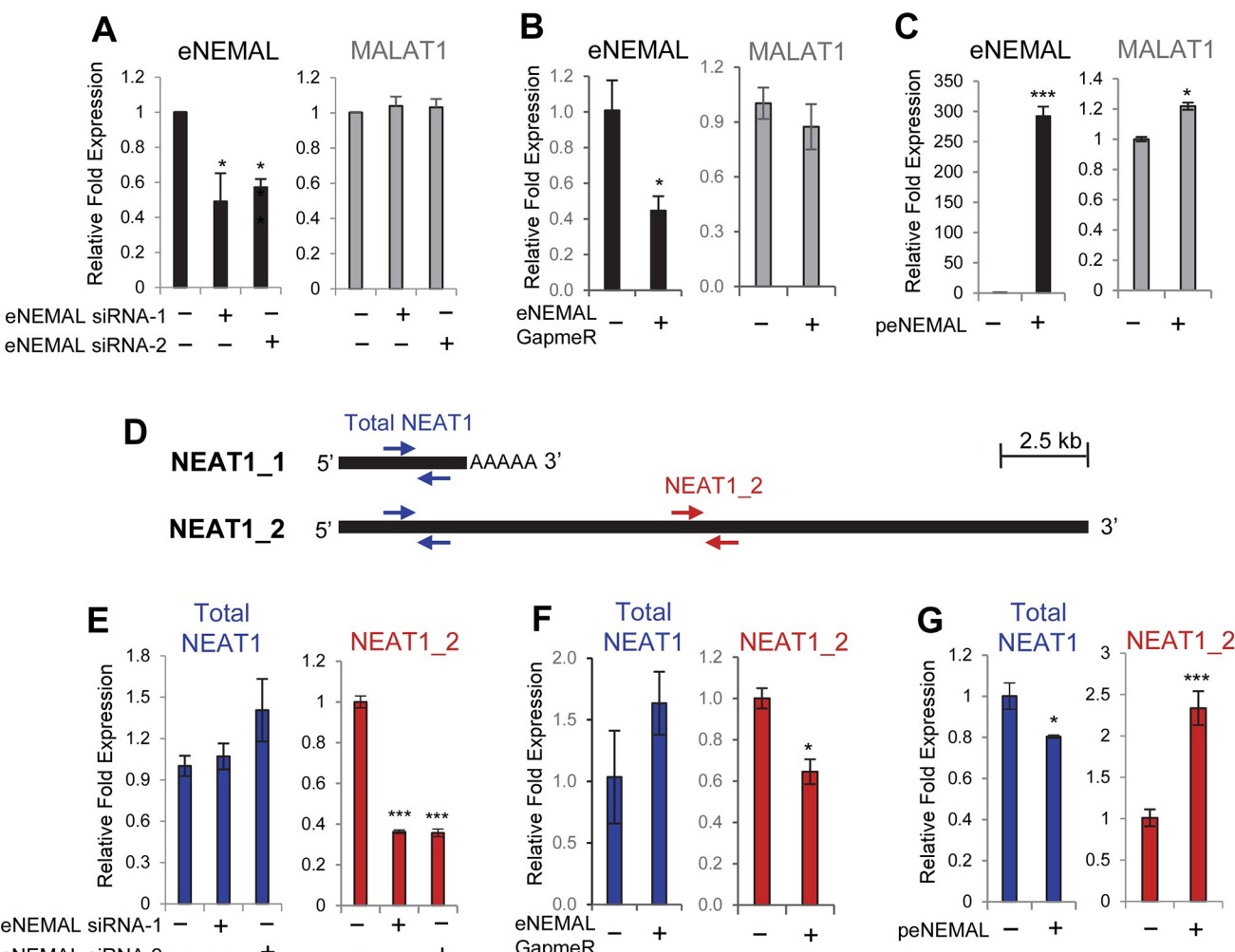

**Fig 5. eNEMAL regulates NEAT1 isoform switching, resulting in upregulation of the NEAT1_2 long isoform in MCF7 breast cancer cells.** (**A, B and C**) RT-qPCR measuring MALAT1 expression after knocking down or overexpressing eNEMAL. MCF cells were transfected with two different siRNAs targeting eNEMAL and maintained under hypoxia (**A**), transfected with GapmeR targeting eNEMAL and maintained under hypoxia (**B**), or transfected with eNEMAL-overexpressing plasmid (pCDH-eNEMAL) and maintained under normoxia (**C**). RNA was isolated 48 hours after siRNA/GapmeR/plasmid transfection and used for RT-qPCR. (**D**) qPCR primer scheme to detect total NEAT1 (both the short isoform NEAT1_1 and the long isoform NEAT1_2; blue arrows) and NEAT1_2 only (red arrows). (**E, F and G**) RT-qPCR measuring total NEAT1 and NEAT1_2. MCF7 cells were transfected with eNEMAL siRNA and maintained under hypoxia (**E**), transfected with GapmeR targeting eNEMAL and maintained under hypoxia (**F**), or transfected with eNEMAL-overexpressing plasmid (pCDH-eNEMAL) and maintained under normoxia (**G**) prior to RNA isolation as described above. Data shown as mean ± SD, n = 4, * $P < 0.01$, *** $P < 0.001$.

## Discussion

One emerging characteristic of active enhancers is transcription of noncoding RNAs by RNA polymerase II [28]. We previously identified three putative enhancers within the *MALAT1* genomic locus which formed chromatin-chromatin interactions in response to hypoxia [13]. Here, we show that a novel eRNA is transcribed under hypoxia from one of these novel enhancers, the -20 kb enhancer, and defined its sequence by performing both 5' and 3' RACE. We named this eRNA "eNEMAL" for its location between the *MALAT1* and *NEAT1* loci and surprisingly found it promotes NEAT1 long isoform expression.

Recently, an RNA-seq variant called global run-on sequencing (GRO-seq) identified nascent lncRNAs expressed in human primary endothelial cells, and the study indicates active

transcription occurs in a region overlapping with the -20 kb enhancer and is increased under hypoxia [29]. Our RACE experiments revealed that eNEMAL was polyadenylated. This was notable as early descriptions of eRNAs isolated from total RNA but not from polyA+ RNA fractions in neuronal tissues [15]. As eRNAs were sequenced from other tissues, subsets have emerged which are often spliced, polyadenylated, have high H3K4me3 enrichment at the promoter, and regulate adjacent genes [5, 30]. Our RACE and whole gene amplification demonstrated that eNEMAL is a single, unspliced exon, but otherwise this transcript matches the features of this latter group of eRNAs.

While we previously found that the -20 kb enhancer is involved in chromatin-chromatin interactions between the *MALAT1* enhancers and the promoter, we found very little effect of the eNEMAL on regulating MALAT1 expression under normoxic or hypoxic conditions. Interestingly, we found that eNEMAL regulates the NEAT1 short/long isoform switch. The long NEAT1_2 isoform is a critical component of paraspeckles, a nuclear sub-compartment which functions in gene regulation [7, 31–33]. Nuclear paraspeckles are notably enriched in breast cancer cells under hypoxia and are important for cellular survival [24]. Furthermore, deletion of NEAT1_1 had no effect on growth or paraspeckle formation, suggesting specific and indispensable function for NEAT1_2 in these processes [34]. It has been shown that regulation of the NEAT1_1 alternative polyadenylation is regulated by several factors, including CPSF6 and NUDT21 which promote NEAT1_1 polyadenylation, and hnRNPK which blocks this polyadenylation process and promotes NEAT1_2 formation [9]. Notably, hnRNPK increases cellular motility and metastasis and is upregulated in receptor-positive breast cancer [35–37]. Interestingly, NEAT1_2 is also upregulated in receptor-positive breast cancer [38]. These data are consistent with our results, which show that eNEMAL expression was highest in receptor-positive breast cancer cell lines, under both normoxic and hypoxic conditions. It is possible that, under stress conditions which favor NEAT1 upregulation and paraspeckle formation such as hypoxia, eNEMAL interacts with hnRNPK or other proteins to prevent alternative polyadenylation and to suppress NEAT1_1 production. Future studies should focus on elucidating the mechanism(s) by which eNEMAL promotes NEAT1_2 generation potentially by inhibiting the polyadenylation process required for the production of NEAT1_1. We also noticed that NEAT1_2 upregulation by overexpression of exogenous eNEMAL was not as efficient as downregulation of NEAT1_2 by eNEMAL siRNA or GapmeR. We speculated that this may be due to the fact that the elevation of local concentration of eNEMAL nearby the place where NEAT1 is transcribed (*i.e.* activation of eNEMAL transcription from its genomic locus) is critical for this eRNA to exert its function on the NEAT1 transcript.

Taken together, our work provides novel insight into the diverse function of eRNAs and suggests that not only may eRNAs play a functional role pre-transcriptionally, but they may also function post-transcriptionally.

## Materials and methods

### Tissue culture and hypoxia

The non-tumorigenic, immortalized mammary cell line MCF10A was obtained from the ATCC and was cultured in MEBM media (Lonza #CC-3150) supplemented with 10% FBS, 1% penicillin/streptomycin, 20 ng/mL EGF, 0.5 mg/mL hydrocortisone, and 10 μg/mL insulin. Breast cancer cell lines BT474, MCF7, MDA-MB-231, MDA-MB-453, SKBR3, and T47D were obtained from the ATCC; MDA-MB-157 and MDA-MB-361 were gifts from Dr. Rob Sobol. All cancer cell lines were maintained in DMEM (Corning #10-017-CV) supplemented with 10% FBS + 1% P/S. All cell lines were maintained in a 37˚C degree incubator with 5% $CO_2$.

Cells were seeded and grown for 48 hours under normoxic (21% O2) conditions then either maintained at normoxia to transferred to BD GasPak™ EZ Gas Generating Systems and Supplies: Pouch System (BD Difco #B260683) for hypoxic (~1% O2) conditions for an additional 24 hours.

## RNA isolation, cDNA synthesis, and qPCR

RNA was extracted using the Qiagen RNeasy Mini Kit (Qiagen #74106), according to the manufacturer's instructions. After RNA quantification, unless otherwise stated, 1 μg of RNA was used for each cDNA synthesis reaction. cDNA was generated using Random Hexamers (Invitrogen #N8080127) and the SuperScript III First Strand Synthesis Kit (Invitrogen #18080051), according to the manufacturer's instructions.

cDNA samples were then treated with RNase H. Quantitative PCR (qPCR) was performed in triplicate reactions using 2X iTaq Universal SYBR Green Supermix (BioRad # 1725121). Primer pairs used are listed in S2 Table.

## 3' Rapid amplification of cDNA ends (RACE)

RNA was extracted by Qiagen RNeasy Mini Kit as described above then 1 μg of RNA was used for cDNA synthesis with an oligo(dT)-Adapter sequence used for priming (5′ –GCTCGCGAG CGCGTTTAAACGCGCACGCGTTTTTTTTTTTTTTTTTTTTVN–3′ ). cDNA products were isolated by QIAquick PCR Purification Kit (Qiagen #28106), according the manufacturer's protocol and eluted in nuclease-free water. PCR I was set up using primers eRNA_3RACE_1F and adapter_R1 (S2 Table). PCR I was run under the following program: 94˚C 5 min, 50˚C 5 min, 72˚C 10 min, followed by 30 cycles of 94˚C 40 sec, 50˚C 1 min, 72˚C 3 min, then a final cycle of 94˚C 40 sec, 50˚C 1 min, and 72˚C 15 min. PCR I products were cleaned by QIAquick PCR Purification Kit (Qiagen #28106) and eluted in nuclease-free water. PCR II was set up using primers eRNA_3RACE_2F and adapter_R2 (S2 Table). The same PCR program was used for PCR II. PCR II amplicons were cloned into the pGEM-T-Vector and sequenced by Sanger sequencing (University of Alabama at Birmingham Hefflin Genomics Center).

## RNA ligase-mediated 3' RACE (3' RLM-RACE)

RNA ligase-mediated 3' RACE (3' RLM-RACE) was performed by the method adopted from Liu *et al*. [39]. Briefly, 200 pmol of the DNA adapter oligo (5′ –ACGCGTGCGCGTTTAAA CGCGCTCGCGAGC– 3′ ) was 5' phosphorylated by T4 Polynucleotide Kinase (T4 PNK, 3' phosphatase minus, NEB #M0236S) per protocol instructions. Next, T4 RNA Ligase 1 (NEB #M0204S) was used to ligate the PNK phosphorylated oligo adapter to 20 pmol total RNA, according to manufacturer's instructions. cDNA was synthesized using the oligo-adaptor-specific primer and nested PCR was performed, similarly to 3' RACE to amplify the oligo adaptor-ligated 3' end of the eRNA. PCR II amplicons were cloned and sequenced as described above.

## 5' RACE

RNA was extracted by Qiagen RNeasy Mini Kit as described above then 2 μg of RNA was used for random hexamer-primed cDNA synthesis. Half of the cDNA was used for a 3' terminal transferase assay (NEB #M0315S), according to the manufacturer's instructions. PCR I was set up using primers eRNA_5RACE_1F and adapter_R1, and oligodT-Adapter (S2 Table). PCR I was run under the following program: 94˚C 5 min, 58˚C 5 min, 72˚C 10 min, followed by 30 cycles of 94˚C 40 sec, 58˚C 1 min, 72˚C 3 min, then a final cycle of 94˚C 40 sec, 58˚C 1 min,

and 72°C 15 min. PCR I products were cleaned by QIAquick PCR Purification Kit and eluted in nuclease-free water. PCR II was set up using primers eRNA_5RACE_2F and adapter_R2 (S2 Table). The same PCR program was used for PCR II, with the exception of each annealing step at 60°C instead of 58°C. PCR II amplicons were screened as above for 3'RACE.

## siRNA and GapmeR transfection

Small interfering RNA (siRNA) were purchased from Sigma Aldrich targeted HIF-1α and HIF-2α. Two different siRNAs for eNEMAL were custom-designed and synthesized by Sigma Aldrich. siRNA was mixed with Opti-MEM media (ThermoFisher), combined with Lipofectamine RNAiMax transfection reagent (Invitrogen) and added to the culture media. siRNA transfection was incubated at 37°C for approximately 8 hours before media change. For HIF siRNA transfection, cultures were grown for 48 hours at normoxia then moved to hypoxia for an additional 24 hours prior to RNA isolation. For eNEMAL siRNA transfection, cultures were grown for 24 hours at normoxia and then moved to hypoxia for an additional 24 hours. siRNA sequences are listed in S3 Table.

Antisense LNA GapmeRs targeting eNEMAL were custom-designed and purchased from Qiagen. For transfection, 40–80 pmol of Negative Control A (Qiagen #339515 LG00000 0020-DDA) and eNEMAL (Qiagen #339511 LG00778462-DDA) GapmeRs were mixed with Opti-MEM media (ThermoFisher), combined with Lipofectamine RNAiMAX transfection reagent (Invitrogen) and added to the culture media, similarly to the siRNA transfection described above. Likewise, the cells were incubated for 24 hours at normoxia then moved to hypoxia for an additional 24 hours prior to RNA isolation. GapmeR sequences are listed in S3 Table.

## Cloning and overexpression of eNEMAL

The eRNA was amplified from MCF7 cDNA (F-AGCCAGGTCCCTTCTTC and R-GCAAAG GTTGTTTTATTAAAAGCTTCC) and cloned into the *Nhe*I and *Bam*HI sites of pCDH-CMV-MCS-EF1-Puro (pCDH-eNEMAL). Constructs were transformed into *E. coli* DH5-α and colonies were selected with X-Gal. Clones were screened for insert with *Hind*III restriction digest and confirmed by sequencing. MCF7 cells were seeded and transfected with either the pCDH-CMV-MCS-EF1-Puro empty vector or pCDH-eNEMAL construct using Lipofectamine 3000 (Invitrogen #L3000008) and Opti-MEM media (Gibco #31985–070). Cells were incubated for 48 hours prior to RNA isolation.

## Statistical analysis

All experiments shown are the result of at least three biological replicates. Data are presented as the mean ± S.D. with $p$ values calculated by 2-tailed $t$ test. All statistics were calculated using GraphPad Prism software version 7.02 (GraphPad).

## Supporting information

**S1 Fig. Sequences identified from oligo(dT)-based conventional 3' RACE.** The final PCR products were loaded in the agarose gel, and the band indicated as #1 and #2 were cut for sequencing. The sequences identified from Sanger sequencing as well as the oligo(dT)-adaptor primer used for cDNA synthesis were shown.
(PDF)

**S2 Fig. Sequences identified from RNA ligase-mediated 3' RACE (3' RLM-RACE).** Products from the final PCR were loaded in the agarose gel, and the band indicated with the red arrow was cut for sequencing. The sequences identified from Sanger sequencing were shown. The 3'

end of the transcript detected by 3' RLM-RACE was located from 84 nt downstream from the cleavage site determined by conventional 3' RACE (blue arrowhead).
(PDF)

**S3 Fig. eNEMAL ortholog alignment.** Alignment generated as described in Fig 3. Accession numbers: *Homo sapiens* MT773342, *Pan troglodytes* NC_036890, *Pan paniscus* CM003394, *Gorilla gorilla* NC_044613, *Pongo abelii* NC_036914, *Nomascus leucogenys* NC_044384, *Macaca fascicularis* NC_022285, *Macaca mulatta* NC_041767, *Macaca nemestrina* KQ00774 5.1, *Papio anubis* NC_044989, *Rhinopithecus bieti* MCGX01000834.1, *Rhinopithecus roxellana* KN295605.1, *Theropithecus gelada* QGDE01000660.1, *Colobus angolensis* KN980607.1, *Piliocolobus tephrosceles* PDMG02000207.1, *Chlorocebus sabaeus* NC_044384, *Mandrillus leucophaeus* KN979072.1, *Aotus nancymaae* KZ200996.1, *Callithrix jacchus* NC_048393, *Cebus capucinus imitator* KV389528.1, and *Saimiri boliviensis* JH378199.1.
(PDF)

**S4 Fig. Upregulation of VEGFA and CA9, well-known hypoxia-induced genes, upon hypoxia.** The indicated cell lines were exposed to hypoxia for 24 hrs and the fold increases of VEGFA and CA9 were calculated (vs. normoxia). Bar graphs show average ± SD, n = 3. VEGFA and CA9 gene expression levels were normalized to the level of YWHAZ expression.
(PDF)

**S5 Fig. Total NEAT1 and the NEAT1 long isoform (NEAT1_2) are differentially regulated in different breast cancer cell lines upon hypoxia, and the high levels of NEAT1_2 induction upon hypoxia were observed in some breast cancer cell lines with the high levels of hypoxia-mediated eNEMAL induction. (A)** Fold increases of total NEAT1 and the long isoform (NEAT1_2) upon hypoxic exposure (24 hrs) determined by RT-qPCR using the primer sets indicated in Fig 5D. Note that total NEAT1 and NEAT1_2 are not always proportionally increased, suggesting that hypoxia leads to alternations in the isoform ratio in addition to NEAT1 transcription. **(B)** Fold increases of NEAT1_2 normalized to fold increases of total NEAT1. Note several breast cancer cell lines with high levels of eNEMAL expression under hypoxia (shown in Fig 4A) show the bigger increase of NEAT1_2 upon hypoxia, suggesting eNEMAL contribution to promoting NEAT1_2 expression. MCF10A, which is a non-tumorigenic line, does not follow the correlation.
(PDF)

**S1 Table. Primate eRNA homology.**
(XLSX)

**S2 Table. Primers used in this study.**
(DOCX)

**S3 Table. siRNA and GapmeR sequences used in this study.**
(DOCX)

**S1 Raw images.**
(PDF)

## Acknowledgments

We thank Dr. Robert Sobol for providing the MDA-MB-157 and MDA-MB-361 cell lines.

## Author Contributions

**Conceptualization:** Joshua K. Stone, Eun-Young Erin Ahn.

**Formal analysis:** Joshua K. Stone, Lana Vukadin.

**Funding acquisition:** Eun-Young Erin Ahn.

**Investigation:** Joshua K. Stone, Lana Vukadin.

**Supervision:** Eun-Young Erin Ahn.

**Writing – original draft:** Joshua K. Stone, Eun-Young Erin Ahn.

**Writing – review & editing:** Joshua K. Stone, Lana Vukadin, Eun-Young Erin Ahn.

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
