## [Decision Letter · Decision Letter 0]

6 Nov 2020

PONE-D-20-27337

eNEMAL, an enhancer RNA transcribed from a distal MALAT1 enhancer, promotes NEAT1 long isoform expression

PLOS ONE

Dear Dr. Ahn,

Thank you for submitting your manuscript to PLOS ONE. After careful consideration, we feel that it has merit but does not fully meet PLOS ONE’s publication criteria as it currently stands. Therefore, we invite you to submit a revised version of the manuscript that addresses the points raised during the review process.

As for the comments of reviewer 1 the authors should repeat the 3’ RACE experiments using a ligation-mediated approach and address the other minor points listed. As suggested by reviewer 2 the authors should try S-oligo or gapmer LNA oligos for NEAT1 knockdown and should check for a variable NEAT1_1/2 ratio in relation to eNEMAl expression.

We look forward to receiving your revised manuscript.

Kind regards,

Massimo Caputi, PhD

Academic Editor

PLOS ONE

Journal Requirements:

Reviewers' comments:

Reviewer's Responses to Questions

**Comments to the Author**

1. Is the manuscript technically sound, and do the data support the conclusions?

Reviewer #1: Partly

Reviewer #2: Partly

2. Has the statistical analysis been performed appropriately and rigorously? 

Reviewer #1: Yes

Reviewer #2: Yes

3. Have the authors made all data underlying the findings in their manuscript fully available?

Reviewer #1: Yes

Reviewer #2: Yes

4. Is the manuscript presented in an intelligible fashion and written in standard English?

Reviewer #1: Yes

Reviewer #2: Yes

5. Review Comments to the Author

Reviewer #1: The authors show that an enhancer element between the MALAT1 and NEAT1 loci is upregulated in multiple cell lines in response to hypoxia. This results in minimal effects on MALAT1 expression, but leads to up-regulation of the long NEAT1 isoform. The manuscript thus suggests a new role for this eRNA in controlling nearby lncRNAs.

(1) Fig 2: The authors use an oligo(dT) based primer for 3’ RACE and conclude that the eRNA is polyadenylated. Many published eRNAs are known to not be polyladenylated and would be missed by the authors’ 3’ RACE approach. Therefore, the authors should repeat the 3’ RACE experiments using a ligation-mediated approach. This will clarify if additional, non-polyadenylated isoforms of the eRNA are expressed in cells.

(2) Figure 2C: Please clarify the identities of the other bands that were observed on the gels shown.

(3) Line 105: The authors write that there was no evidence for a plus strand eRNA by qPCR or RACE, but signal for such a transcript was observed by RNA-seq in Figure 1B. The authors need to provide an explanation why different results have been observed.

(4) Fig S1 is difficult to interpret as presented. The authors write on line 107 that this region is highly conserved, so it would for example be helpful to provide the percent identity between different sets of species.

(5) Fig 3A: To better understand why there are differences in eNEMAL expression with hypoxia in different cell lines, it would be helpful to provide additional qPCR data showing that known hypoxia-induced genes are induced in all of the cell lines shown.

(6) Fig 3C: Please discuss why HIF2A levels change with HIF1A is knocked down.

Minor comments

(1) Line 67: I would caution against using subjective terms like “strong”. Compared to MALAT1, the expression of the eRNA is very, very weak.

(2) Line 153: Be careful with language. eNEMAL is only upregulated in some cell lines in response to hypoxia.

Reviewer #2: In this manuscript, the authors characterized a novel lncRNA named eNEMAL, which is transcribed from an enhancer located between MALAT1 and NEAT1. The transcription start/termination sites of eNEMAL were determined by RACE PCR, and they found the expression of eNEMAL was induced upon hypoxia in multiple cancerous cell lines. They also performed functional analyses of eNEMAL using siRNA, and found that knockdown of eNEMAL lead to isoform switch of Neat1;i.e., increased amount of Neat1_1 at the expense of NEAT1_2. Conversely, overexpression of eNEMAL lead to the induction of NEAT1_2. The authors propose eNEMAL regulates isoform switch of NEAT1 by sequestering certain factors necessary for the alternative polyadenylation to produce NEAT1_1.

Their findings, especially Fig4D and 4E, are very interesting and provide a new insight into the function of enhancer derived RNAs. However, from a stoichiometric point of view, I am not convinced if 50% reduction of eNEMAL is enough to induce such a robust change of NEAT1_1/2 ratio given the number of abundant RNA binding proteins such as hnRNPK or CPSF6, which usually far exceed the number of lncRNAs in the nucleus. More plausible explanation is that these siRNA somehow target NEAT1_2-specific regions (off-target effects) and thus specifically knockdown NEAT1_2 leaving NEAT1_1 intact. Additional experiments should be required to further confirm the quantitative correlation between the amount of eNEMAL and the ratio of NEAT1_1/2.

Major points

1. In general, siRNA knockdown is not efficient for nuclear lncRNAs due to the nuclear localization of Ago proteins. They should try S-oligo or gapmer LNA oligos for NEAT1 knockdown, which usually enables 80-90% decrease of target nuclear lncRNAs. The more efficient knockdown should lead to the more prominent effect on the NEAT1 isoform switch.

2. They observed differential expression of eNEMAL in various cell lines (Figure 3A). If eNEMAL really regulate NEAT1 isoform switch, they should observe variable NEAT1_1/2 ratio that correlate with the expression of eNEMAL. This should be checked.

Minor points

Citations are somewhat biased and should be modified appropriately.

1. Sunwoo et al (Genome Res 19:347-) should be cited for essential role of NEAT1_2 during paraspeckle formation in addition to ref 8 & 9.

2. Li et al (RNA 23: 872-) and Isobe et al (RNA 26: 251-) should be cited for the role of NEAT1_1 during paraspeckles formation in addition to ref 31

6. PLOS authors have the option to publish the peer review history of their article (what does this mean?). If published, this will include your full peer review and any attached files.

Reviewer #1: No

Reviewer #2: No

---

## [Author Response · Author response to Decision Letter 0]

22 Apr 2021

Responses to Reviewers

RE: PONE-D-20-27337

eNEMAL, an enhancer RNA transcribed from a distal MALAT1 enhancer, promotes NEAT1 long isoform expression

We thank the editor and the reviewers for their interest in our work as well as their constructive comments and suggestions. Since we had difficulties in getting the research materials and reagent in a timely manner due to the pandemic, we requested the extension of the revision time. We sincerely appreciate your understanding. Owing to your understanding and support, we were able to complete the experiments to improve our manuscript and made revisions accordingly. We hope these revisions satisfy the reviewer’s concerns and have strengthened our manuscript. 

Overall comments from the Editor

As for the comments of reviewer 1 the authors should repeat the 3’ RACE experiments using a ligation-mediated approach and address the other minor points listed. As suggested by reviewer 2 the authors should try S-oligo or gapmer LNA oligos for NEAT1 knockdown and should check for a variable NEAT1_1/2 ratio in relation to eNEMAL expression.

As described below, we have performed ligation-mediated 3’ RACE (RNA ligase-mediated 3’ RACE; 3’RLM-RACE) experiments and repeated the conventional oligo(dT)-adaptor-mediated 3’ RACE. We presented these data in Figure 2 and Supplemental Figures S1-2. We also used GapmeR LNA oligos to knockdown the eRNA eNEMAL and measured the expression of both total NEAT1 and NEAT1_2, confirming our previous findings. The results are now included in Figure 5. 

Reviewer #1

(1) Fig 2: The authors use an oligo(dT) based primer for 3’ RACE and conclude that the eRNA is polyadenylated. Many published eRNAs are known to not be polyladenylated and would be missed by the authors’ 3’ RACE approach. Therefore, the authors should repeat the 3’ RACE experiments using a ligation-mediated approach. This will clarify if additional, non-polyadenylated isoforms of the eRNA are expressed in cells.

We thank the review for this valuable suggestion. We performed RNA ligase-mediated 3’ RACE (3’ RLM-RACE) by ligating an adaptor oligo to the free 3’ end of total RNA using T4 RNA ligase, followed by cDNA synthesis using an adaptor oligo-specific primer, nested PCR and cut-band sequencing. Interestingly, we detected a transcript which contains additional 84 nt when compared to the transcript detected by oligo(dT)-adaptor-mediated conventional 3’ RACE. We included the data in revised Figure 2, Supporting information (S2 Fig.). At this point, it is not clear that this transcript is one of the read-through products or another major form of the eRNA without a poly(A) tail. We discussed this point in the revised manuscript. Nevertheless, we were able to reproducibly detect the 3’ end at the +762 position by conventional 3’ RACE, confirming the presence of the poly(A)-tailed eRNA eNEMAL. 

(2) Figure 2C: Please clarify the identities of the other bands that were observed on the gels shown.

To address this point, we repeated 3’ RACE (conventional 3’ RACE) multiple times, further optimized the PCR condition to eliminate non-specific PCR amplification, and found the reproducible band pattern (3’ RACE DNA gel in Figure 2C). From sequencing, the top band was identified as the eRNA of our interest (eNEMAL, chromosomal location 11q13.1), confirming the 3’ end sequence of eNEMAL. The bottom band was identified as part of the cathepsin D (CTSD) gene, which is located at the 11p15.5 position (S1 Fig). This result confirm that the bottom band is a non-specific band amplified by our 3’RACE condition. 

(3) Line 105: The authors write that there was no evidence for a plus strand eRNA by qPCR or RACE, but signal for such a transcript was observed by RNA-seq in Figure 1B. The authors need to provide an explanation why different results have been observed.

We found that we did not provide clear descriptions on Figure 1B, which shows the transcription levels (RNA-seq) as well as histone marks at the -20 kb enhancer region. We used the different “data ranges” for the plus-strand (0-24) and the minus-strand (0-138) to visualize peaks from the RNA-seq data. Therefore, the transcription level of plus-strand is significantly low compared to that of the minus-strand. We have added this note to the figure legend. 

(4) Fig S1 is difficult to interpret as presented. The authors write on line 107 that this region is highly conserved, so it would for example be helpful to provide the percent identity between different sets of species.

We acknowledge this was poorly worded and presented on our part. To correct this, we first clarified that the NEAT1-MALAT1 locus is highly conserved in terms of gene synteny in chordates. Next, we added both a heatmap of percent nucleotide homology of each eRNA ortholog to the human eNEMAL and each other (new Figure 3) and provided the values as source data in New S1 Table (Excel file). We believe this additional numerical data helps provide context and eases interpretation of the sequence alignment.

(5) Fig 3A: To better understand why there are differences in eNEMAL expression with hypoxia in different cell lines, it would be helpful to provide additional qPCR data showing that known hypoxia-induced genes are induced in all of the cell lines shown.

This is a valuable point. We performed qPCR to measure the expression level of VEGFA and CA9, well-known hypoxia-induced genes, and included the data in Supporting information (S4 Fig). We were able to detect upregulation of VEGFA and CA9 upon hypoxia in all the cell lines used. However, the degree of upregulation varied in each cell line and VEGFA, CA9 and eNEMAL were not proportionally upregulated under hypoxia. We speculate that although hypoxia indeed induces the expression of numerous target genes, including newly identified eNEMAL, the levels/degrees of upregulation of each gene vary in different cell lines. This could be due to unique mutations and different expression levels of other factors that also influence the expression of these target genes. 

(6) Fig 3C: Please discuss why HIF2A levels change with HIF1A is knocked down.

As the reviewer noticed, we observed that HIF2A is increased when HIF1A siRNA was transfected. It has been shown that HIF1A function as a repressor of HIF-2A expression in MCF7 cells (Stiehl DP et al. Oncogene (2012) 31:2283). We indicated this point in the figure legend abd cited the associated reference. 

Minor comments

(1) Line 67: I would caution against using subjective terms like “strong”. Compared to MALAT1, the expression of the eRNA is very, very weak.

We agree with the reviewer and modified our description. 

(2) Line 153: Be careful with language. eNEMAL is only upregulated in some cell lines in response to hypoxia.

We made changes to accurately describe our findings. We indicated that our finding about eNEMAL regulation of NEAT1 isoform expression was observed in MCF7 cells. 

Reviewer #2: 

Major points

1. In general, siRNA knockdown is not efficient for nuclear lncRNAs due to the nuclear localization of Ago proteins. They should try S-oligo or gapmer LNA oligos for NEAT1 knockdown, which usually enables 80-90% decrease of target nuclear lncRNAs. The more efficient knockdown should lead to the more prominent effect on the NEAT1 isoform switch.

The reviewer raised a valuable point (We believe the reviewer meant “S-oligo or gapmer LNA oligos for eNEMAL knockdown”, not NEAT1 knockdown). To verify the effect of eNEMAL knockdown on NEAT1 isoform expression, we used GapmeR LNA oligo targeting eNEMAL and were able to decrease eNEMAL by about 60%. In this condition, we were also able to detect the long isoform of NEAT1 (NEAT1_2) is downregulated while the total NEAT1 level did not show statistically significant change. We designed and tested 3 more GapmeRs with target sequences, however, those GapmeR failed to decrease eNEMAL. It is likely that potential secondary structure of eRNA interferes with the targeting process by siRNA or GapmeR. Although we could not achieve 80-90% decrease of eNEMAL by GapmeR or siRNA, all experiments using 2 different siRNAs and a GapmeR showed consistent results, supporting our conclusion. These data have been added to new Figure 5 and the manuscript has been revised accordingly. 

2. They observed differential expression of eNEMAL in various cell lines (Figure 3A). If eNEMAL really regulate NEAT1 isoform switch, they should observe variable NEAT1_1/2 ratio that correlate with the expression of eNEMAL. This should be checked 

As the reviewer suggested, we measured how much total NEAT1 and the long isoform (NEAT1_2) are increased when various cell lines were exposed to hypoxia. Interestingly, we found that total NEAT1 and NEAT1_2 were not proportionally increased upon hypoxia. We found that several breast cancer cell lines with high levels of eNEMAL expression under hypoxia (shown in Fig 4A) show the bigger increase of NEAT1_2 upon hypoxia, suggesting eNEMAL contribution to promoting NEAT1_2 production. We included the data in Supporting Information (new S5 Fig) and discussed these finding in the Results section. 

Minor points

Citations are somewhat biased and should be modified appropriately.

1. Sunwoo et al (Genome Res 19:347-) should be cited for essential role of NEAT1_2 during paraspeckle formation in addition to ref 8 & 9.

Thank you for this critical suggestion. We have reviewed the referred paper and cited it. 

2. Li et al (RNA 23: 872-) and Isobe et al (RNA 26: 251-) should be cited for the role of NEAT1_1 during paraspeckles formation in addition to ref 31.

We have reviewed these papers and cited item accordingly. We appreciate the reviewer’s advice on these references.

---

## [Decision Letter · Decision Letter 1]

28 Apr 2021

eNEMAL, an enhancer RNA transcribed from a distal MALAT1 enhancer, promotes NEAT1 long isoform expression

PONE-D-20-27337R1

Dear Dr. Ahn,

We’re pleased to inform you that your manuscript has been judged scientifically suitable for publication and will be formally accepted for publication once it meets all outstanding technical requirements.

Kind regards,

Massimo Caputi, PhD

Academic Editor

PLOS ONE

Additional Editor Comments (optional):

Reviewers' comments:

Reviewer's Responses to Questions

**Comments to the Author**

1. If the authors have adequately addressed your comments raised in a previous round of review and you feel that this manuscript is now acceptable for publication, you may indicate that here to bypass the “Comments to the Author” section, enter your conflict of interest statement in the “Confidential to Editor” section, and submit your "Accept" recommendation.

Reviewer #1: (No Response)

Reviewer #2: All comments have been addressed

2. Is the manuscript technically sound, and do the data support the conclusions?

Reviewer #1: Yes

Reviewer #2: Yes

3. Has the statistical analysis been performed appropriately and rigorously? 

Reviewer #1: Yes

Reviewer #2: Yes

4. Have the authors made all data underlying the findings in their manuscript fully available?

Reviewer #1: Yes

Reviewer #2: Yes

5. Is the manuscript presented in an intelligible fashion and written in standard English?

Reviewer #1: Yes

Reviewer #2: Yes

6. Review Comments to the Author

Reviewer #1: The authors have addressed my concerns.

One comment: Given that different results were obtained with the two 3’ RACE approaches, I suggest the authors to be more careful when describing the eRNA as polyadenylated (Line 51, Line 91, Line 244). It would be more accurate to say, for example, that “it may be polyadenylated” or that at least a portion of it is polyadenylated. I am surprised the polyadenylated isoform was not detected with the ligation-based approach, but this may be due to PCR conditions used.

Reviewer #2: (No Response)

7. PLOS authors have the option to publish the peer review history of their article (what does this mean?). If published, this will include your full peer review and any attached files.

Reviewer #1: No

Reviewer #2: No

---

## [Editor Report · Acceptance letter]

14 May 2021

PONE-D-20-27337R1 

eNEMAL, an enhancer RNA transcribed from a distal *MALAT1* enhancer promotes NEAT1 long isoform expression 

Dear Dr. Ahn:

I'm pleased to inform you that your manuscript has been deemed suitable for publication in PLOS ONE. Congratulations! Your manuscript is now with our production department. 

Kind regards, 

on behalf of

Dr. Massimo Caputi 

Academic Editor

PLOS ONE